# Altered NKp46 Recognition and Elimination of Influenza B Viruses

**DOI:** 10.3390/v13010034

**Published:** 2020-12-27

**Authors:** Alexandra Duev-Cohen, Batya Isaacson, Orit Berhani, Yoav Charpak-Amikam, Nehemya Friedman, Yaron Drori, Michal Mandelboim, Ofer Mandelboim

**Affiliations:** 1The Concern Foundation Laboratories at the Lautenberg Center for Immunology and Cancer Research, The Hebrew University Hadassah Medical School, Jerusalem 9112001, Israel; cohen.alexandra@mail.huji.ac.il (A.D.-C.); batya.isaacson@mail.huji.ac.il (B.I.); orit.berhani@mail.huji.ac.il (O.B.); Yoav.Amikam@mail.huji.ac.il (Y.C.-A.); 2Central Virology Laboratory, Ministry of Health, Public Health Services, Chaim Sheba Medical Center, Tel Hashomer, Ramat-Gan 5265601, Israel; Nehemya.Friedman@sheba.health.gov.il (N.F.); Yaron.Drori@sheba.health.gov.il (Y.D.); michalman@sheba.health.gov.il (M.M.)

**Keywords:** NK cells, NKp46, NCR1, influenza B, hemagglutinin

## Abstract

Every year, millions of people worldwide are infected with influenza, causing enormous health and economic problems. The most common type of influenza is influenza A. It is known that Natural Killer (NK) cells play an important role in controlling influenza A infection, mostly through the recognition of the viral protein hemagglutinin (HA) by the activating receptor, NKp46. In contrast, little is known regarding NK cell recognition of influenza B viruses, even though they are responsible for a third of all pediatric influenza deaths and are therefore included in the seasonal vaccine each year. Here we show that NKp46 also recognizes influenza B viruses. We show that NKp46 binds the HA protein of influenza B in a sialic acid-dependent manner, and identified the glycosylated residue in NKp46, which is critical for this interaction. We discovered that this interaction has a binding affinity approximately seven times lower than NKp46 binding of influenza A’s HA. Finally, we demonstrated, using mice deficient for the mouse orthologue of NKp46, named NCR1, that NKp46 is not important for influenza B elimination. These findings enable us to better understand the interactions between the different influenza viruses and NK cells that are known to be crucial for viral elimination.

## 1. Introduction

Natural Killer (NK) cells are lymphocytes that are part of the innate immune system [1]. They are critical in the defense against virus-infected cells, bacteria, parasites, fungi, and malignant cells [2]. NK cell activity is regulated by the balance of signals received from inhibitory and activating receptors. The Natural Cytotoxicity Receptors (NCRs), NKp30, NKp44, and NKp46, are among the main activating receptors expressed by NK cells. NKp30 and NKp46 are constitutively expressed, whereas NKp44 is upregulated after NK cell activation [3]. NKp46 recognizes the hemagglutinin (HA) protein of various viruses including influenza A (IAV) [4], poxvirus [5], Sendai virus [4], and Newcastle disease virus [6]. It is also known to bind fungal ligands, such as the EPA adhesins on Candida glabrata [7], and an unknown bacterial ligand of Fusobacterium nucleatum [8]. NKp46 has also been shown to bind the endogenous ligands heparan sulfate [9] and complement factor P [10], but it still has additional unknown tumor [11] and cellular ligands [12,13].

Influenza infections can lead to hospitalization and death among high-risk individuals (for example, elderly and chronically ill patients), thus posing major health and economic burdens. Part of IAV’s deadly nature is its ability to change at a high frequency the virus’ membrane proteins, hemagglutinin (HA) and neuraminidase (NA) [14], a phenomenon known as antigenic shift. Accordingly, new influenza strains to which the world population is not immune arise every year. However, the ability of HA to bind sialic acids is conserved throughout the different strains, since this function facilitates influenza binding and infection of human cells [15]. NK cells take advantage of this property to recognize and kill IAV infected cells. The glycosylated threonine 225 residue on NKp46 interacts with HA and thus activates NK cell function to kill the infected cells [16]. The mouse orthologue of NKp46, called NCR1, also recognizes HA via the same mechanism. This has been shown using mice deficient for Ncr1, in which the influenza infection displays increased lethality [17].

Influenza B (IBV) is somewhat less common than IAV and almost exclusively infects humans. The only two known lineages of IBV were discovered in the late 1970s and are named Victoria and Yamagata [18]. Despite the fact that it mutates at a significantly slower rate than IAV [19], IBV is still responsible for a third of all pediatric influenza deaths and is included in the seasonal vaccine each year [18].

Here we show that NKp46 recognizes IBV via binding to HA. We discovered the mechanism of binding and identified the binding site on NKp46 that is crucial for its interaction with IBV’s HA. We found that the affinity of NKp46 to the HA of IBV is significantly weaker than to that of IAV and that NKp46 is not involved in the clearance of IBV in vivo.

## 2. Materials and Methods

### 2.1. Cells and Viruses

The cell lines used in this study were the human choriocarcinoma cell line JEG-3 and the murine lymphoma cell line EL4 (ATCC). Human peripheral blood lymphocytes were separated from buffy coats using centrifugation on Ficoll gradient (Lymphoprep) followed by NK cells’ isolation using the Easy Sep Negative selection human NK cells enrichment kit (StemCell Technologies) according to manufacturer’s protocol. Mouse NK cells were isolated from the immune competent Ncr1+/gfp and from the NCR1-deficient Ncr1gfp/gfp mice using the Easy Sep Negative selection mouse NK cells’ enrichment kit (StemCell Technologies) 18 h after intraperitoneal injection of 200 μg poly(I):poly(C) (Sigma). The human influenza viruses A/Brisbane/59/2007 H1N1 and B/Brisbane/60/2008 (Victoria lineage) used in this study were generated as previously described [20].

### 2.2. Antibodies, Fusion Proteins, and Compounds

Monoclonal antibodies (mAbs) used in the present study included anti-HA specific to A/Brisbane/59/2007 and anti-HA specific to B/Brisbane/60/2008 that we kindly received from the International Reagent Resource (IRR) that was established by the Center for Disease Control and Prevention (CDC). The hNKp46.02 antibody was previously described [21]. The anti-NKp46 (9E2) was purchased from Biolegend. The generation of the fusion proteins NKp46-Ig, T225A-Ig, and NCR1-Ig was previously described. The purity of the fusion proteins was near 100%. Treatment of NKp46-Ig with Neuraminidase (NA) beads (Sigma, St. Louis, MO, USA) was performed as previously described [22]. For the ELISA experiments, we used the following recombinant HA proteins fused to an HIS-Tag: HA1 (A/Brisbane/59/2007), HA3 (A/Wisconsin/12/2010), and HA Victoria (B/Brisbane/60/2008) that we kindly received from the International Reagent Resource (IRR) that was established by the CDC.

### 2.3. FACS Staining of Viruses

Cells were coated overnight with 20 μL of A/Brisbane/59/2007 H1N1 or B/Brisbane/60/2008 at 37 °C. The cells were then washed and incubated with the appropriate antibody (0.2 μg/well) or fusion protein (0.5 μg to 5 μg/well). Then, cells were stained and analyzed using fluorescence-activated cell sorting (FACS). All results were analyzed using the FCSExpress 6 software.

### 2.4. ELISA

The various recombinant HA proteins mentioned above were coated to an ELISA plate (0.1 μg/well). Blocking was performed with BSA 1%. 0.1 μg/well of NKp46-Ig or anti-HIS-Tag (clone AD1.1.10 from Bio-Rad, Kidlington, Oxford, England) were added followed by the relevant biotinylated secondary antibody from Jackson ImmunoResearch. The plate was then incubated with Streptavidin HRP (Jackson ImmunoResearch, West Grove, PA, USA). Binding was measured by using TMB solution and reading of the results was done with an ELISA reader at a wavelength of 650 nm. The binding of NKp46 to the different HA proteins was normalized to the amount of HA that was bound to the well, which we determined using the anti HIS-tag antibody.

### 2.5. Surface Plasmon Resonance (SPR)

We used HIS capture kit (GE Healthcare, Piscataway, NJ, USA)to coat the different recombinant HA proteins to a CM5 chip (GE Healthcare). After coating of the HA, we performed single cycle kinetics, as described before [23], by pushing different concentrations of NKp46-Ig.

### 2.6. Killing Assay

Targets with or without influenza were incubated overnight with radioactive ^35^S labeled methionine. In the same time, human activated NK cells were incubated with hNKp46.02 for the internalization assay, as previously described [21]. The next day, the human NK cells or the mouse NK cells were incubated for 5 h at 37 °C with the targets and the cytotoxic activity was measured by ^35^S release from lysed cells, as previously described [24].

### 2.7. In Vivo Influenza Infections

All animal work was conducted according to relevant national and international guidelines. The work was approved by the Hebrew University Medical School Ethic committee (Ethics number MD-19-15748-3, date of approval 7/7/19). All experiments were performed using 6–8- weeks-old females of C57BL/6 background. The generation of the Ncr1gfp/gfp mice was described before [25]. Mice were intranasally inoculated with 40µL of A/Brisbane/59/2007 H1N1 or B/Brisbane/60/2008 (Victoria lineage). Five days post infection, lungs were harvested and viral load was assessed.

### 2.8. Statistical Methods

GraphPad Prism software version 8.0 was used for all statistical analyses. For statistical significance, two-tailed Student’s t test analysis was used. A statistical test was considered significant (*) when *p* < 0.05. ANOVA was used to identify significant group differences See figure legend for further detail.

## 3. Results

### 3.1. NKp46 Binds IBV

To test whether NK cells recognize IBV, we incubated Jeg3 cells with IAV or IBV. Jeg3 cells express very few NK activating ligands and are therefore not killed well by human NK cells, making them ideal candidates for testing the interaction between NK cells and the abovementioned viruses [26]. For IAV we used the H1N1 virus A/Brisbane/59/2007 viral strain and for IBV we used the B/Brisbane/60/2008 viral strain which belongs to the Victoria lineage. Using anti-HA antibodies specific to either IAV or IBV, we verified that both viruses bind Jeg3 cells (Figure 1a). Next, we generated a fusion protein composed of the extracellular portion of human NKp46 fused to the Fc portion of human IgG1 (named NKp46-Ig) and stained the Jeg3 cells in the presence or absence of the viruses. As we previously reported [4], increased NKp46-Ig staining was observed against Jeg3 cells in the presence of IAV (Figure 1b,c data in red). Similarly, we observed increased binding of NKp46-Ig to IBV coated cells (Figure 1b,d data in blue ). Collectively, we showed that NKp46 recognizes not only IAV (as previously shown, [4]), but also IBV.

### 3.2. NKp46 Has Lower Affinity to the HA of IBV than IAV

We previously demonstrated that the viral ligand of IAV recognized by NKp46 is the HA protein [4], so we inferred that NKp46 binds IBV in a similar manner. Initially, we examined binding of NKp46 to the virally specific HAs by ELISA assays using recombinant HIS-tagged HA proteins obtained from the International Reagent Resource (IRR). ELISA plates were coated with the different HAs and then incubated with the NKp46-Ig fusion protein. As a negative control we used the HA of the H3N2 virus from 2010, which is known not to bind NKp46 [27]. As can be seen in Figure 2a, NKp46 did not bind the HA of the H3N2 virus (middle bar, black dots) as expected. However, interestingly, NKp46 bound both to the HA of IBV (blue dots) and to the HA of IAV (red dots). To further investigate the NKp46 binding, we measured the binding affinities of NKp46 to the virally specific HAs. We used surface plasmon resonance (SPR) where we coated a CM5 chip with anti-HIS antibodies and then bound the HA recombinant proteins of IAV H1N1 (Figure 2b) or IBV Victoria (Figure 2c). Using single cycle kinetics, we then injected NKp46-Ig in increasing concentrations and measured the dissociation constant, KD. The affinity of NKp46 to IBV was seven times less than the affinity to IAV (Figure 2c).

### 3.3. Threonine 225 Is Important for the Binding of NKp46 to IBV

Since NKp46 binds the HA of IBV with lower affinity than IAV, we next tested whether the recognition of IBV by NKp46 is different from that of IAV. We previously showed that the binding of NKp46 to IAV is dependent on sialic acids present on NKp46 [22]. To test whether the binding of NKp46 to IBV is also dependent on sialic acids we treated NKp46-Ig with neuraminidase, an enzyme known to cleave sialic acid residues [28]. We incubated the NKp46-Ig fusion protein with neuraminidase beads (NA beads) for 2 h at 37 °C. Then, we stained Jeg3 cells in the presence or absence of IBV using the treated NKp46-Ig. A significant decrease in the binding of NKp46-Ig was observed following treatment of the fusion proteins with NA (Figure 3a), indicating that the binding of NKp46 to IBV is sialic acid-dependent.

The specific glycosylated residue critical for the binding of NKp46 to IAV was previously identified to be threonine 225 (Thr 225) [16]. To test whether threonine 225 is also involved in the recognition of the IBV HA, we generated a fusion protein of NKp46 in which we mutated Thr 225 into alanine (T225A-Ig). We then stained Jeg3 cells in the presence of IBV with NKp46-Ig and T225A-Ig. We observed a significant decrease in the binding to IBV when we used the mutant T225A-Ig as compared to NKp46-Ig (Figure 3b). These findings indicate that sialylation and Thr 225 play an important role in NKp46 recognition of IBV, but other unknown elements in NKp46 exist that are essential for NKp46 recognition of IBV.

### 3.4. Human NK Cells Require NKp46 to Efficiently Eliminate IBV

To investigate the functional implications of NKp46 binding to IBV we used an anti-NKp46 mAb, named hNKp46.02, that we previously generated. This specific antibody, when incubated with cells expressing NKp46, leads to internalization and degradation of the receptor; therefore, NKp46 is downregulated from the cell surface [21].

We incubated bulk primary human NK cells with an isotype control or with hNKp46.02 for 16 h at 37 °C. As we previously reported [21], such incubation resulted in 50% reduction of NKp46 surface expression due to NKp46 internalization (Figure 4a). These NK cells were subsequently incubated with ^35^S-labeled Jeg3 cells in the presence or absence of IBV and a cytotoxicity assay was performed. In the absence of IBV, Jeg3 cells were weakly killed and the hNKp46.02 antibody reduced the killing even further (although, not in a statistically significant manner, Figure 4b). When Jeg3 cells were incubated with IBV, increased killing was observed. The increased killing was dependent on NKp46, since killing was reduced to baseline levels of uninfected Jeg3 cells when NK cells were treated with hNKp46.02 (Figure 4b).

### 3.5. The Mouse Orthologue of NKp46 Recognizes IBV but Does not Mediate Killing by Mouse NK Cells

We next wanted to determine whether the NKp46 mouse orthologue, NCR1, also recognizes IBV. For this we incubated IBV with EL4 cells, a murine lymphoma cell line, and verified binding using anti-HAs’ antibodies (Figure 5a). We then used a previously generated fusion protein composed of NCR1 fused to human IgG1, called NCR1-Ig [25], and saw that NCR1-Ig binds both IBV (Figure 5b,d data in blue) and IAV (Figure 5b,c data in red).

To understand the significance of the binding we performed killing assays with murine NK cells extracted from the immune competent, Ncr1+/gfp (HET), mice or from the Ncr1 deficient, Ncr1gfp/gfp (KO), mice that we previously generated [25]. In the NCR1 KO mouse the Ncr1 gene was replaced with a green fluorescent protein (GFP). In the HET immune competent mice one allele of Ncr1 was still present and the other allele was replaced by GFP. Hence, all NK cells were labeled with GFP. The presence of GFP in the NK cells enables a reliable and clean isolation of murine NK cells.

We incubated the NK cells obtained from the two strains of mice with EL4 in the presence or absence of IAV and IBV. As expected, and in agreement with previous results [17], when we used Ncr1+/gfp (HET) NK cells that express the NCR1 receptor, we observed increased killing of IAV (Figure 5e). This increased killing was dependent on NCR1 because in its absence no difference was observed when IAV was present or absent (Figure 5e). Moreover, NK cells isolated from HET and KO mice killed uninfected EL4 to a similar extent, showing no general effect of reduced cytotoxic ability. However, when the cells were incubated with IBV, no significant increased killing was observed in the HET mice NK similarly to the KO mice NK cells (Figure 5e), indicating that NCR1 may not play a role in the killing of IBV in mice.

### 3.6. NKp46 Is Not Involved in the Killing of IBV In Vivo

To determine the physiological relevance of our findings we injected mice intranasally with IAV or IBV. Five days post infection we sacrificed the mice and assessed viral load in the lungs. As previously reported [17], when we infected mice with IAV we observed that Ncr1gfp/gfp (KO) exhibits severely impaired viral clearance, thus indicating that the elimination of IAV is dependent on NCR1 (Figure 6a). Importantly, when the mice were injected with IBV, no difference in viral clearance between Ncr1gfp/gfp and Ncr1+/+ mice was observed (Figure 6b). These results suggest that the lower affinity binding of NKp46 to the HA of IBV might not be enough to activate the NK cells in vivo.

## 4. Discussion

Influenza infections constitute a major health and economic burden [29]. According to the World Health Organization (WHO), up to 650,000 people die of influenza every year. To evade recognition by the immune system, IAV viruses undergo rapid changes, in particular in their HA and NA proteins. This high mutation rate is one of the causes for the inability of the adaptive immune system to develop memory against this virus. In this process however, the ability of HA to bind sialic acid remains unchanged. This property is crucial for the virus to attach and infect the cell [30]. One of the major NK receptors, NKp46, uses the sialylated threonine 225 residue to bind the HA protein of IAV [16]. Accordingly, the virus evolved and developed mechanisms to evade recognition by NK cells. The virus uses its NA enzyme to cleave the sialic acids present on NKp46, thus preventing recognition of the infected cells [22]. Furthermore, recent H3N2 isolates display lower sialic acid affinity due to a mutation in the glycosylation on the HA protein, and, indeed, NKp46 is unable to recognize these viruses [27]. It is a continuous and unrelenting battle between our NK cells and the influenza virus family.

Interestingly, it was not known whether NK cells recognize IBV. Influenza B viruses diverged from influenza A viruses at some point in the distant past and are, therefore, newer viruses [18]. This virus mutates at a significantly lower rate than IAV and is composed of only one serotype and two lineages, Yamagata and Victoria [31]. Although not as deadly as IAV, IBV is the reported cause of death of one-third of all infected children every year, as stated by the WHO. IBV is more dangerous to children due to age-dependent sialic acid composition. Sialic acid composition in the human respiratory tract changes during aging with higher α-2,3-linked sialic acids’ expression in children and more α-2,6-linked sialic acids in adults. Interestingly, while IAV HA binds to 2,6-linked sialic acids, IBV HA can bind both α-2,3 and α-2,6-linked sialic acids. Therefore, children are more susceptible to infection by IBV [32].

Here we showed that NKp46 binds to the HA of IBV mainly through sialic acids present on the threonine 225 of NKp46. This mode of binding is similar to that of IAV [16]. Interestingly, the presence of sialic acids, per se, is not sufficient for binding to HA as most NK cell receptors express sialic acids, but we previously showed that most NK cell receptors are not able to interact with HA [33]. Moreover, when we stained cells in the presence of IBV with the mutated T225A-Ig we could still observe some binding to the influenza virus. Thus, there are elements other than the presence of sialic acids which determine the specificity of NK receptor-HA interactions.

Next, we demonstrated that NKp46 has a lower affinity against the HA of IBV as compared to IAV. Nevertheless, the NK cells were still able to kill cells in the presence of IBV, and this was dependent on NKp46. Interestingly, when assessing whether mouse NK cells were able to kill cells in the presence of IBV, we observed that, although NCR1 is able to bind IBV, this interaction does not induce killing. In this assay both IAV and IBV were evaluated, and mouse NK cells expressing NCR1 were able to efficiently kill cells in the presence of IAV but not IBV. This observation suggests that NCR1 may not play a role in the clearance of IBV. We confirmed this assessment using an in vivo system where we observed that the presence or absence of NCR1 did not influence viral clearance of IBV. It is possible that the lower affinity of NKp46 to IBV HA is because the HA protein of IBV binds α-2,3 as well as α-2,6-linked sialic acids. The sialic acids present on the threonine 225 of NKp46 are mainly α-2,6-linked sialic acids [34]. Moreover, it has been shown that IBV, in general, has lower receptor binding affinities than influenza A viruses due to critical difference in the sialic acid binding site of HA, namely a Phe-95 versus Tyr-98 [35]. This difference influences the conformation of the protein and might explain the limited host range of influenza B viruses [32].

## Figures and Tables

**Figure 1 viruses-13-00034-f001:**
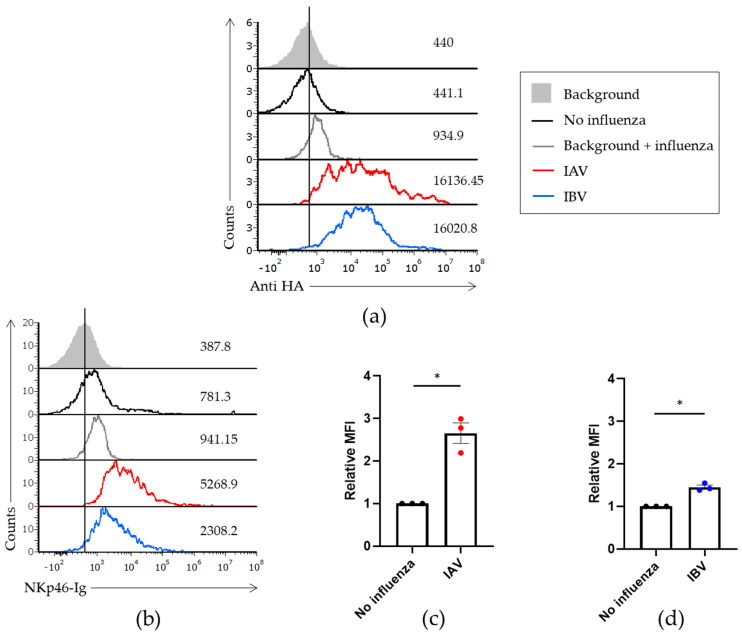
NKp46 binds IAV and IBV. Jeg3 cells were incubated or not (black histogram) with A/Brisbane/59/2007 H1N1 (IAV) or B/Brisbane/60/2008 (IBV). (**a**) FACS staining of IAV (red histogram) and IBV (blue histogram) with antibodies against the hemagglutinin of IAV and IBV, respectively. (**b**) The cells incubated with the indicated viruses were stained with NKp46-Ig fusion protein. The filled histograms and gray, empty histograms represent the staining with secondary antibodies only in the absence or presence of influenza, respectively. Figure shows one representative experiment out of three performed. (**c**,**d**) MFI quantification of three independent experiments including data presented in (**b**). FACS staining with NKp46-Ig against IAV (**c**) and IBV (**d**) was normalized to the staining of the cells without influenza. Values are shown as mean ± SEM. * *p* < 0.05 by two-tailed paired Student’s *t* test.

**Figure 2 viruses-13-00034-f002:**
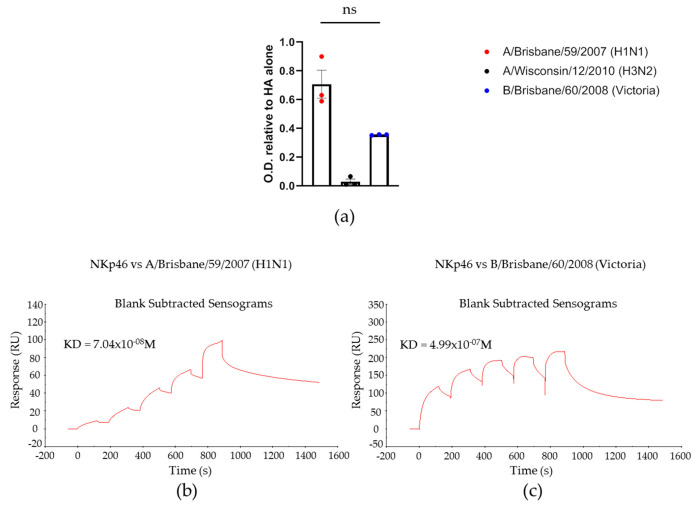
NKp46 binds the hemagglutinin protein of IAV better than IBV. (**a**) Plates were coated with the indicated recombinant HIS-tag labeled hemagglutinin proteins, incubated or not with NKp46-Ig and assayed for binding using ELISA. The results present the NKp46-Ig binding relative to the amount of HA coated on the plate (determined using an anti HIS-tag antibody). An ANOVA test was performed in order to evaluate significance. Figure shows cumulative results of three independent experiments. (**b**,**c**) SPR single cycle kinetics of NKp46 vs. the recombinant hemagglutinin proteins of IAV (**b**) and IBV (**c**). CM5 chip was coated with anti-HIS-tag antibodies and then bound with the recombinant HAs expressing HIS-tag. NKp46-Ig was injected with the highest concentration of 6.6 μM in the left panel and 48.2 μM in the right panel.

**Figure 3 viruses-13-00034-f003:**
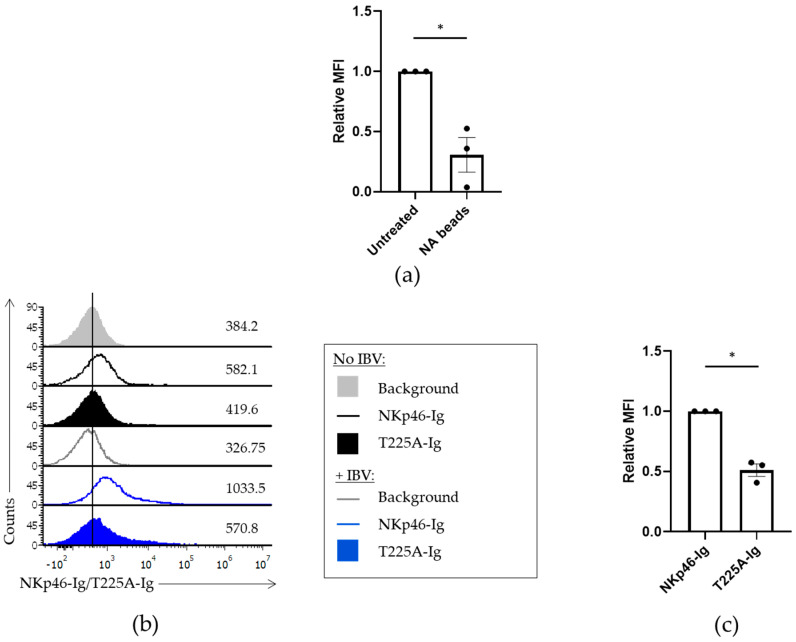
The binding of NKp46 to IBV is dependent on the sialic acid present on threonine 225. (**a**) NKp46-Ig fusion proteins were treated or not with Neuraminidase beads (NA beads) and then incubated with Jeg3 cells in the presence of IBV. The graphs summarize average of three independent experiments. The results were normalized to staining of the cells with untreated NKp46-Ig. Values are shown as mean ± SEM. * *p* < 0.05 by two-tailed paired Student’s *t* test. (**b**) Jeg3 incubated or not with IBV was stained with NKp46-Ig (No IBV = black histogram; +IBV = blue histogram) or the mutant NKp46 T225A-Ig (No IBV = black, filled histogram; +IBV = blue, filled histogram). The gray, filled histograms and gray, empty histograms represent the staining with secondary antibodies only in the absence or presence of influenza, respectively. Figure shows one representative experiment out of three performed. (**c**) MFI quantification of three independent experiments including data presented in (**b**). FACS staining with T225A-Ig against IBV was normalized to the staining of the cells with NKp46-Ig. Values are shown as mean ± SEM. * *p* < 0.05 by two-tailed paired Student’s *t* test.

**Figure 4 viruses-13-00034-f004:**
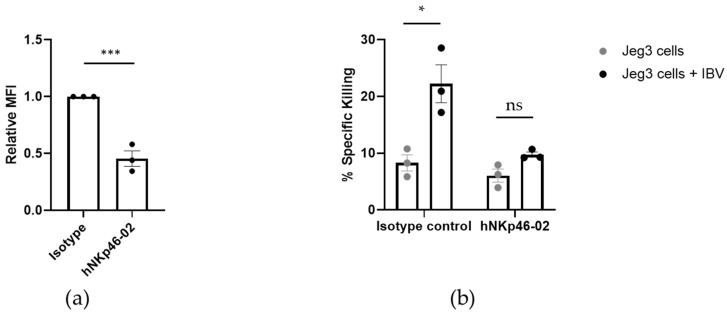
When expression of NKp46 is reduced from the cell surface, NK cells are not able to effectively kill IBV. NK cells were incubated with hNKp46.02 internalizing antibody for 16 h at 37 °C. (**a**) Reduction of the expression of NKp46 receptor from the cell surface was verified by FACS staining. The Median Fluorescence Intensity (MFI) of the staining is presented. The graphs summarize average of three independent experiments. The results were normalized to staining of the NK cells incubated with an isotype control. Values are shown as mean ± SEM. *** *p* < 0.001 by two-tailed paired Student’s *t* test. (**b**) NK cells treated (shown in (**a**)) with an isotype control or with the internalizing hNKp46.02 antibody were incubated with Jeg3 cells in the presence (black dots) or absence (gray dots) of IBV. Killing assays were performed for 5 h. An ANOVA test was performed in order to evaluate significance. Values are shown as mean ± SEM. * *p* < 0.05. Figure shows cumulative results of three independent experiments.

**Figure 5 viruses-13-00034-f005:**
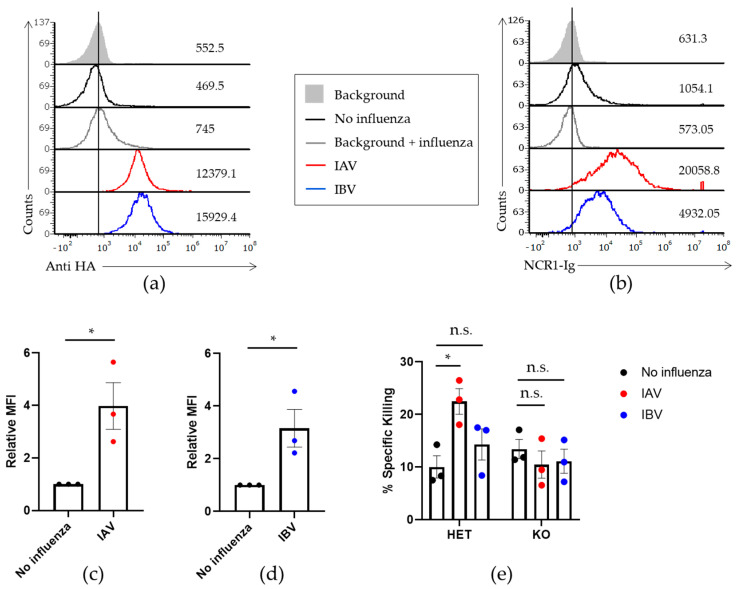
NCR1 binds to IBV but does not activate killing by mouse NK cells. (**a**,**b**) EL4 cells were incubated or not (black histogram) with IAV or IBV. (**a**) FACS staining of IAV (red histogram) and IBV (blue histogram) with antibodies against the hemagglutinin of IAV and IBV, respectively. (**b**) The cells incubated with the indicated viruses were stained with NKp46-Ig fusion protein. The filled histograms and gray empty histograms represent the staining with secondary antibodies only in the absence or presence of influenza, respectively. Figure shows one representative experiment out of three performed. (**c**,**d**) MFI quantification of three independent experiments including data presented in (**b**). FACS staining with NCR1-Ig against IAV (**c**) and IBV (**d**) was normalized to the staining of the cells without influenza. Values are shown as mean ± SEM. * *p* < 0.05 by two-tailed paired Student’s *t* test. (**e**) NK cells from Ncr1+/gfp (HET) and Ncr1gfp/gfp (KO) mice were extracted and incubated with EL4 cells in the presence or absence of IAV and IBV. Killing assay was performed as mentioned above. An ANOVA test was performed in order to evaluate significance. Values are shown as mean ± SEM. * *p* < 0.05. Figure shows cumulative results of three independent experiments.

**Figure 6 viruses-13-00034-f006:**
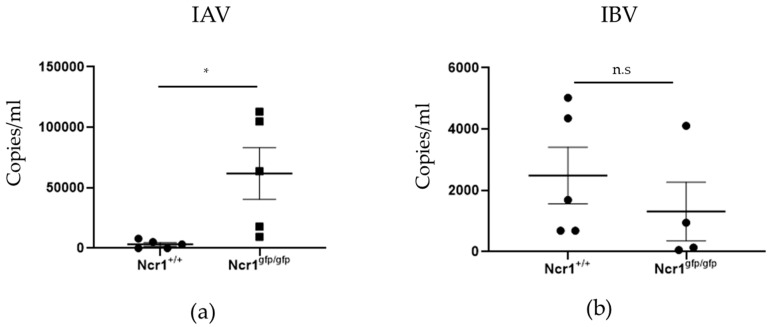
NCR1 is critical for the control of IAV but not IBV infection in vivo. Ncr1+/+ and Ncr1gfp/gfp mice were intranasally infected with IAV (**a**) or IBV (**b**). After five days, lungs were collected and real-time PCR was performed. n = 5 mice in each group. Values are shown as mean ± SEM. * *p* < 0.05 by two-tailed Student’s *t* test.

## Data Availability

Data is contained within the article.

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
