# Peer review of "Altered NKp46 Recognition and Elimination of Influenza B Viruses"

_viruses, 2020, doi:10.3390/v13010034_

Round 1

Reviewer 1 Report

The authors describe several experiments that show the role of NKp46 in influenza B infection using in vitro and mouse-infected models. Although the presentation of the experiments’ sequence and the scientific logic are perfect, there are vital points that need to be addressed in a revised version of the manuscript. 

  1. No statistical section in the method is presented. However, the authors mentioned statistical significance but without any proper statistical methodology. 
  2. All figures showing histograms of the binding should also show cumulative values of binding for the three experiments with proper statistical analysis. For instance, Fig 1b shows NKp46-Ig binding, but the values are somewhat close to the background and influenza condition. Therefore, a proper statistical analysis should be performed. Similarly, the same thing should be applied to the data in Fig 3b and Fig 5b. 
  3. In the results headings, the authors wrote, “NK cell killing of IBV is dependent on NKp46,” and then in the following section, they wrote “Mouse NK cells do not effectively kill IBV even though NCR1 binds IBV”. Although the two approaches are different, these subtitles are confusing and need to be adjusted according to the experimental approach that has been done. 
  4. Similar to point number 3 above, the authors mentioned in the discussion, “Next, we demonstrated that NKp46 has a lower affinity against the HA of IBV as compared to IAV. This lower affinity did not seem to impair the in vitro NKp46 dependent killing of cells in the presence of IBV. However, even though we could observe binding of IBV by NCR1, mice NK cells were not able to kill IBV in vitro suggesting that NCR1 may not play a role in the clearance of IBV.” This discrepancy could be due to using human cell lines versus murine cell lines. Therefore, caution should be addressed when describing and discussing the results. 

Minor points:

  1. The manuscript should be proofread. For instance, in legends of figures 2, 4, and 5, the authors used at the beginning of the sentence the word “shown”, which is not appropriate. 
  2. Figure 5C has no y-axis title.  

Author Response

The authors describe several experiments that show the role of NKp46 in influenza B infection using in vitro and mouse-infected models. Although the presentation of the experiments’ sequence and the scientific logic are perfect, there are vital points that need to be addressed in a revised version of the manuscript. 

We thank the reviewer for the critical reading of our manuscript and for the points raised. We responded to criticisms and amended the manuscript accordingly (changes are in track changes).   

  1. No statistical section in the method is presented. However, the authors mentioned statistical significance but without any proper statistical methodology.

We thank the reviewer for pointing this out. We added a ‘statistical methods’ section in the ‘Materials and Methods. Please see 2.8 lines 115-118. 

  1. All figures showing histograms of the binding should also show cumulative values of binding for the three experiments with proper statistical analysis. For instance, Fig 1b shows NKp46-Ig binding, but the values are somewhat close to the background and influenza condition. Therefore, a proper statistical analysis should be performed. Similarly, the same thing should be applied to the data in Fig 3b and Fig 5b. 

We agree with the reviewer’s comment and added cumulative results of 3 independent experiments to Figure 1b (called Figure 1c), to Figure 3b (called Figure 3c) and to Figure 5b (called Figure 5c)

  1. In the results headings, the authors wrote, “NK cell killing of IBV is dependent on NKp46,” and then in the following section, they wrote “Mouse NK cells do not effectively kill IBV even though NCR1 binds IBV”. Although the two approaches are different, these subtitles are confusing and need to be adjusted according to the experimental approach that has been done. 
  1. Based on the reviewer comment we have changed the titles accordingly:

“Human NK cells require NKp46 to efficiently eliminate IBV” (line 199)

“The mouse ortholog of NKp46 recognizes IBV but does not mediate killing by mouse NK cells” (line 226)

We also modified the Figure 4 legend’s title to “When expression of NKp46 is reduced from the cell surface, NK cells are not able to effectively kill IBV” (lines 214-215)

  1. Similar to point number 3 above, the authors mentioned in the discussion, “Next, we demonstrated that NKp46 has a lower affinity against the HA of IBV as compared to IAV. This lower affinity did not seem to impair the in vitro NKp46 dependent killing of cells in the presence of IBV. However, even though we could observe binding of IBV by NCR1, mice NK cells were not able to kill IBV in vitro suggesting that NCR1 may not play a role in the clearance of IBV.” This discrepancy could be due to using human cell lines versus murine cell lines. Therefore, caution should be addressed when describing and discussing the results.

Thank you for this comment. We have rewritten this section to make our explanation clearer. “Next, we demonstrated that NKp46 has a lower affinity against the HA of IBV as compared to IAV. Nevertheless, the NK cells were still able to kill cells in the presence of IBV, and this was dependent on NKp46. Interestingly, when assessing whether mouse NK cells were able to kill cells in the presence of IBV, we observed that although NCR1 is able to bind IBV, this interaction does not induce killing. In this assay both IAV and IBV were evaluated, and mouse NK cells expressing NCR1 were able to efficiently kill cells in the presence of IAV but not IBV. This observation suggests that NCR1 may not play a role in the clearance of IBV.” (lines 306-313).

We used IAV as a control knowing that killing of IAV by mouse NK cells is dependent on NCR1. We can then exclude the possibility that murine cells versus human cells can interfere with the recognition of IBV by the NK cells.

Minor points:

  1. The manuscript should be proofread. For instance, in legends of figures 2, 4, and 5, the authors used at the beginning of the sentence the word “shown”, which is not appropriate.

We corrected the manuscript and removed the word ‘shown’ from the beginning of sentences. Instead, we wrote ‘Values are shown’ (for example in line 142) 

  1. Figure 5C has no y-axis title. 

Thank you for pointing this out. The figure has been modified to show the cumulative results of 3 experiments and it includes a y-axis title. Please see figure 5d.

Reviewer 2 Report

This paper demonstrates binding of Influenza B virus (IBV) haemagglutinin (HA) to NKp46 and analyses the functional consequences for viral control in a mouse model. The authors demonstrate lower affinity interaction between NkP46 and IBV HA in comparison to its binding to Influenza A virus (IAV) HA. The data are presented with clarity, are appropriately controlled and the conclusions are supported by the data. The data presented point towards relatively subtle differences between NKp46 mediated recognition of IBV and IAV viruses.

The following points should be addressed:

  1. The author should show summary data with mean values for all three experiments represented in Figure 2a (not just replicates of within a single experiment). What is the range of KD for NKp46 binding to IAV and IBV in the three experiments as shown in Figures 2b and 2c.
  2. Please also show the summary data for all three experiments represented in figures 3 and 4 and 5.
  3. Please insert the Y-axis label in figure 5c
  4. Do subsets of the primary human NK cells (CD56bright vs CD56dim for example) bind HA from IBV or IAV with different intensity according to NKp46 expression levels? Which subsets are involved in the killing of IBV infected Jeg3 cells, for example.
  5. The authors should consider alteration of the title to ‘Altered NKp46 recognition…’ or ‘Reduced NKp46 recognition….’ As opposed to ‘Defective…’ as this better reflects the data. Additionally it cannot be assumed that the binding pattern for HAI is the archetypal scenario, considering the reduced genetic variability of IBV. The last paragraph of the discussion should be toned down as it is a big jump to attributed distinct pathologies of IAV and IBV in human to NKp46 interactions.

Author Response

Reviewer 2

This paper demonstrates binding of Influenza B virus (IBV) haemagglutinin (HA) to NKp46 and analyses the functional consequences for viral control in a mouse model. The authors demonstrate lower affinity interaction between NkP46 and IBV HA in comparison to its binding to Influenza A virus (IAV) HA. The data are presented with clarity, are appropriately controlled and the conclusions are supported by the data. The data presented point towards relatively subtle differences between NKp46 mediated recognition of IBV and IAV viruses.

We thank the reviewer for the critical reading of our manuscript and for the points raised. We responded to criticisms and amended the manuscript accordingly (changes are in track changes).   

The following points should be addressed:

  1. The author should show summary data with mean values for all three experiments represented in Figure 2a (not just replicates of within a single experiment).

Thank you for your comment. All the figures have been modified to include cumulative results of 3 experiments. For stainings we added a graph of the cumulative results (Figure 1c, 3c and 5c). For the other experiments we modified the graphs so they include summary of all three experiments (Figure 2a, Figure 3a, Figure 4a and 4b and Figure 5d)

  1. What is the range of KD for NKp46 binding to IAV and IBV in the three experiments as shown in Figures 2b and 2c.

The experiments shown in Figure 2b and 2c were performed only once due to lack of reagents to repeat the experiments (mainly the recombinant HA proteins). If the reviewer thinks we should repeat these experiments we will obtain more reagents and do that.  

  1. Please also show the summary data for all three experiments represented in figures 3 and 4 and 5.

We corrected the figures as we wrote before (please see the first point).

  1. Please insert the Y-axis label in figure 5c

Thank you for pointing this out. The figure has been modified to show the cumulative results of 3 experiments and it includes a y-axis title. Please see figure 5d.

  1. Do subsets of the primary human NK cells (CD56bright vs CD56dim for example) bind HA from IBV or IAV with different intensity according to NKp46 expression levels? Which subsets are involved in the killing of IBV infected Jeg3 cells, for example.

This is an interesting point that we did not investigate. We were giving few days to correct the manuscript. We will investigate this issue in the future. However, if the reviewer thinks that this experiment is critical we will be happy to do that.

The NK cells used in all of the experiments are activated primary human CD56dim cells.

  1. The authors should consider alteration of the title to ‘Altered NKp46 recognition…’ or ‘Reduced NKp46 recognition….’ As opposed to ‘Defective…’ as this better reflects the data.

We agreed with the reviewer and changed the title to ‘Altered NKp46 recognition and elimination of influenza B viruses (line 2).

Additionally it cannot be assumed that the binding pattern for HAI is the archetypal scenario, considering the reduced genetic variability of IBV. The last paragraph of the discussion should be toned down as it is a big jump to attributed distinct pathologies of IAV and IBV in human to NKp46 interactions.

Following the reviewer comment we removed the last paragraph from the manuscript.

Round 2

Reviewer 1 Report

The choice of Student t test for 3 independent experiments is not accurate. If there are more than two groups, one way ANOVA with a post hos test should be used, or a non-parametric such as Kruskal Wallis followed by Dunn's as a post test should be applied.  

Author Response

The choice of Student t test for 3 independent experiments is not accurate. If there are more than two groups, one way ANOVA with a post hos test should be used, or a non-parametric such as Kruskal Wallis followed by Dunn’s as a post test should be applied.

We thank the reviewer for this comment and we corrected accordingly (previous changes are marked in yellow, current changes are in track changes). Please see the Statistical Method section (lines 115-119) and various figure legends.

In the FACS staining experiments, since the comparison is done independently between IAV and no influenza samples and IBV and no influenza samples, we do not compare more than two groups. Thus, two tailed paired T test was performed and the data presented accordingly. To make the comparisons clearer, we separated the graph into two different graphs (please see Figures 1c,1d and Figures 5c,5d and elaborated Figure legend).

As the reviewer rightfully suggested, for experiments where more than two groups were compared a non-parametric one-way ANOVA was performed with Dunn’s correction for multiple comparisons (Figures 2a, 4b and 4e and corresponding legends). The appropriate changes were mentioned in the text as well (please see lines 151-153).